# Zero-Shot Continuous Prompt Transfer: Generalizing Task Semantics Across Language Models

**Zijun Wu**[1], **Yongkang Wu**[2], **Lili Mou**[1,3]
[1]Dept. Computing Science & Alberta Machine Intelligence Institute (Amii), University of Alberta
[2]Huawei Poisson Lab    [3]Canada CIFAR AI Chair
`zijun4@ualberta.ca, wuyongkang7@huawei.com,`
`doublepower.mou@gmail.com`

## Abstract

Prompt tuning in natural language processing (NLP) has become an increasingly popular method for adapting large language models to specific tasks. However, the transferability of these prompts, especially continuous prompts, between different models remains a challenge. In this work, we propose a zero-shot continuous prompt transfer method, where source prompts are encoded into a relative space and the corresponding target prompts are searched for transferring to target models. Experimental results confirm the effectiveness of our method, showing that "task semantics" in continuous prompts can be generalized across various language models. Moreover, we find that combining "task semantics" from multiple source models can further enhance the performance of transfer.[1]

## 1 Introduction

Recently, natural language processing (NLP) has witnessed a paradigm shift from the finetuning of full language models to the optimization of a small subset of prompt tokens (Shin et al., 2020; Lester et al., 2021; Li & Liang, 2021; Zhong et al., 2021). As language models have dramatically increased in size and may contain billions of parameters (Brown et al., 2020), the strategy of freezing language models while optimizing the learnable prompt parameters becomes the most affordable and efficient alternative for downstream tasks. This technique, referred to as prompt tuning, has gained substantial recognition for its effectiveness across a range of language models (Shin et al., 2020; Lester et al., 2021; Li & Liang, 2021; Zhong et al., 2021).

Various prompt tuning methods have been explored, which can be generally categorized into discrete and continuous cases. Discrete prompt tuning, such as AutoPrompt (Shin et al., 2020), primarily focuses on the selection and optimization of a predetermined set of tokens within a language model's vocabulary. By contrast, continuous prompt tuning (Zhong et al., 2021) allows the modification of continuous prompt embeddings by gradient descent. The latter typically offers better performance on downstream tasks due to its greater flexibility in the prompt space. However, existing prompt tuning often requires accessing the model's internal states, as the gradient needs to be backpropagated to the first layer of token embeddings (Shin et al., 2020; Zhong et al., 2021), which contradicts the goal of avoiding gradient computation for large language models.

Therefore, it would be ideal if we can perform prompt tuning on a small model (which is computationally inexpensive) and transfer the prompt to large models. We thus question: How transferable are these prompts between different language models? Prior research on transferring prompts mainly focuses on discrete prompts (Rakotonirina et al., 2023). Such transfer is often straightforward, as discrete prompt tokens usually carry semantic meanings by their nature and can be directly accepted by different language models. For continuous prompts, however, prompt transfer becomes less straightforward because they are unexplainable and sparsely distributed in a high-dimensional space (Khashabi et al., 2022; Su et al., 2022). Moreover, different models might learn the embedding

---

[1]Our code is available at `https://github.com/MANGA-UOFA/PTfer`

space differently, due to their designs, sizes, training paradigms, as well as parameter random initializations. Therefore, transferring a continuous prompt tailored for one model to another, especially with different dimensions, remains a challenge.

Attempts to bridge this gap have centered around introducing a neural projector that aligns continuous prompts across different models. However, the learned projectors are specific to unique model pairs. More importantly, it introduces extra computational cost because the training requires task supervision on the target model and the source prompt embeddings, or even the need to utilize the parallel prompt embeddings for both models (Su et al., 2022). These approaches cannot be applied in a zero-shot transfer scenario, and are undesired in real applications.

In this work, we propose a novel approach to zero-shot continuous prompt transfer without the need for task supervision or additional training of neural projectors. We introduce an encode-then-search strategy, where we encode the source prompts into a relative space (Norelli et al., 2022; Moschella et al., 2023) and then search for the corresponding target prompt embeddings. Our intuition is that the induced continuous prompt contains implicit information for a task (Vu et al., 2022; Wang et al., 2022), referred to as "task semantics", which may be carried over from the source embedding space to the target. We suggest that, although direct transfer of prompt embeddings is problematic because different language models have their own embedding spaces, the position of the continuous prompt embedding relative to the embeddings of known words is more likely to share the same structure in different language models, inspired by the evidence of representation learning literature in other domains, such as word embeddings (Faruqui & Dyer, 2014; Lazaridou et al., 2015; Artetxe et al., 2018), synthetic structure discovery (Wu et al., 2023), unsupervised neural translation (Lample et al., 2018), and cognitive science (Levakov et al., 2021; Chersoni et al., 2021). In our approach, the transfer of prompts only requires a shared vocabulary of common tokens, which serve as the anchors of the relative embedding space. For the target model, we search for its prompt embeddings that preserve the same relative structure as that of the source language model.

Our experiments confirm that, with our proposed zero-shot approach, continuous prompts are transferable to different language models, largely outperforming baseline approaches such as training neural projectors. We also discover that utilizing continuous prompts from multiple distinct source models enhances the generalizability on target models. This is because the semantics of the prompts induced from a single source might be model-specific, whereas the multi-source method provides a more robust view of the task semantics, therefore achieving higher performance of prompt transfer. In short, our contributions are summarized as follows:

- We address a novel setting of zero-shot continuous prompt transfer, which allows for the reuse of continuous prompts across different language models.
- We propose an encode-then-search strategy that maps a continuous prompt into a relative space for transfer between language models. Our approach facilitates multi-source transfer, which cannot be easily done by previous work.
- We provide detailed experimental analysis on a factual-probing suite of 41 types of questions to show the effectiveness of our approach.

## 2 METHODOLOGY

In this section, we begin with an overview of continuous prompt tuning in §2.1. We then introduce our encoding (§2.2) and decoding (§2.3) methods for transferring continuous prompt embeddings between language models. Finally in §2.4, we discuss our multi-source transfer approach for improving the performance of transfer.

### 2.1 CONTINUOUS PROMPT TUNING

Continuous prompt tuning (Zhong et al., 2021) optimizes the embeddings of a prompt in the continuous space for a downstream task, which essentially introduces virtual tokens to the language model. During this process, the continuous prompt, which is a set of learnable embedding vectors, can capture the implicit information for the task of interest (Vu et al., 2022). Different from full-model finetuning methods (Zhou & Srikumar, 2022), the gradient from the final loss function backpropagates through the model but only updates the initial embedding layers.

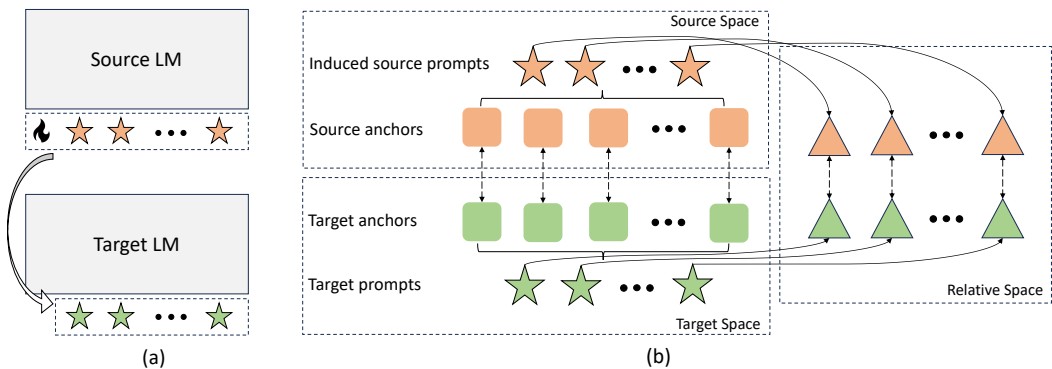

Figure 1: (a) The goal of transferring the induced continuous prompts on a source model to a target model. (b) Our proposed method for this transfer in a zero-shot manner, where the target prompts should be aligned with the induced source prompts in the relative space.

Consider prompting a language model for some task. A continuous prompt has the following format:

$$\texttt{Prompt}(x) = \texttt{x} \ \texttt{v}_1 \ \texttt{v}_2 \ \cdots \ \texttt{v}_m \tag{1}$$

where $x$ is an input data sample, and $m$ is a pre-defined prompt length, i.e., the number of learnable vectors. The configuration of $\texttt{v}_i$ as either a prefix or postfix to $x$ is an aspect of prompt design. In our implementation, we append these tokens as a postfix to $x$. For each virtual token $\texttt{v}_i$, its embedding is a learnable vector $\boldsymbol{v}_i \in \mathbb{R}^d$ that has the same dimension $d$ as the embedding layer of the language model. The soft prompt tuning objective is to maximize the likelihood of the output $y$ of the training sample, given by the source model $P_s(\cdot)$ as

$$\underset{\boldsymbol{v}_1, \cdots, \boldsymbol{v}_m}{\arg \max} \sum_{(x,y) \in D} \log P(y \mid \boldsymbol{v}_1, \cdots, \boldsymbol{v}_m, x) \tag{2}$$

After the continuous prompts $\boldsymbol{v}_1, \ldots, \boldsymbol{v}_m$ are optimized, they are usually used to make inference on the same model for the same task (Lester et al., 2021; Li & Liang, 2021; Zhong et al., 2021).

Prompt tuning is more efficient than full-model finetuning because only a small number of parameters, i.e., the embeddings of the virtual tokens, are updated for a downstream task. Meanwhile, it maintains substantial flexibility and achieves similar performance to full-model finetuning (Lester et al., 2021). However, learning these virtual tokens may still be expensive for large language models because we need to perform backpropagation through the entire model structure. Our goal is to explore the feasibility of learning a continuous prompt with a small language model and transferring it to larger ones, therefore avoiding excessive gradient computations of prompt tuning for different large language models.

## 2.2 Encoding to a Relative Space

We propose to transfer continuous prompts from a source language model to target models. The process involves two key phases: (1) encoding the source prompt embeddings into a relative representation and (2) searching for target prompt embeddings whose corresponding relative representation aligns with those of the source. This two-phase method facilitates the effective transfer of task semantics between different embedding spaces, allowing the transferred prompts to accomplish the task on the target model.

The concept of relative representation involves encoding a data point based on its similarities to certain reference points, known as anchors (Norelli et al., 2022; Moschella et al., 2023). In our work, the relative space, where these encoded representations reside, serves as a shared semantic space across different models and facilitates the transfer process of continuous prompts.

Consider a continuous prompt $\boldsymbol{v}_1, \boldsymbol{v}_2, \cdots, \boldsymbol{v}_m \in \mathbb{R}^{d_s}$, where $d_s$ indicates the embedding dimension of the source language model. We aim to transform it into a relative representation. This can be shown in Figure 1b as transferring orange stars to orange triangles.

To encode the relative embeddings of the prompt, we need a set of common tokens, serving as anchors, that are shared between both source and target language models. We simply choose the

shared tokens as the set of anchors, as they provide a common ground for expressing a prompt in relative terms, regardless of differently learned embedding spaces. Specifically, the anchors' embeddings in the source model can be represented by a matrix $\boldsymbol{A}^{\mathrm{s}} = [\boldsymbol{a}_1^{\mathrm{s}}, \boldsymbol{a}_2^{\mathrm{s}}, \cdots, \boldsymbol{a}_k^{\mathrm{s}}] \in \mathbb{R}^{d_{\mathrm{s}} \times k}$, where $k$ is the number of anchors, and $[,]$ concatenates column vectors into a matrix.

We then encode a prompt embedding $\boldsymbol{v}_i$ in a relative space with respect to these anchors, where we compute the cosine similarity between a prompt embedding and each anchor's embedding, given by

$$\boldsymbol{r}_{\boldsymbol{A}^{\mathrm{s}}}(\boldsymbol{v}_i) = (\cos(\boldsymbol{v}_i, \boldsymbol{a}_1^{\mathrm{s}}), \cdots, \cos(\boldsymbol{v}_i, \boldsymbol{a}_k^{\mathrm{s}}))^{\top} \tag{3}$$

This encoding step translates a continuous prompt from the source language model into a language using the relationship among common tokens so that other models can potentially understand. It bridges the source and target language models and passes implicit task information contained in the source continuous prompt.

## 2.3 SEARCH IN THE TARGET SPACE

We search a continuous prompt for the target language model, based on the intuition that the relative embeddings are model-agnostic and can be aligned across different language models, i.e., the orange and green triangles in Figure 1 should have the same structure. In this way, we can search the (absolute) target prompt embeddings by maximizing the alignment of the source and target relative spaces.

Concretely, the target embeddings $\boldsymbol{v}_1^{\mathrm{t}}, \boldsymbol{v}_2^{\mathrm{t}}, \cdots, \boldsymbol{v}_m^{\mathrm{t}} \in \mathbb{R}^{d_{\mathrm{t}}}$ are randomly initialized, where $d_{\mathrm{t}}$ is the target embedding dimension and may not be the same as $d_{\mathrm{s}}$. They are represented by green stars in Figure 1b. These target embeddings are then encoded using the target anchor embeddings $\boldsymbol{A}^{\mathrm{t}} = [\boldsymbol{a}_1^{\mathrm{t}}, \boldsymbol{a}_2^{\mathrm{t}}, \cdots, \boldsymbol{a}_k^{\mathrm{t}}] \in \mathbb{R}^{d_{\mathrm{t}} \times k}$, shown by green squares in Figure 1b. These target anchors are the same as source anchors, and their embeddings are given by the target language model. Similar to encoding the source prompt, the target embeddings can be represented in the relative space by

$$\boldsymbol{r}_{\boldsymbol{A}^{\mathrm{t}}}(\boldsymbol{v}_i^{\mathrm{t}}) = (\cos(\boldsymbol{v}_i^{\mathrm{t}}, \boldsymbol{a}_1^{\mathrm{t}}), \cdots, \cos(\boldsymbol{v}_i^{\mathrm{t}}, \boldsymbol{a}_k^{\mathrm{t}}))^{\top} \tag{4}$$

To align source and target relative embeddings, i.e., Eqns. (3) and (4), we seek a target embedding $\boldsymbol{v}_i^{\mathrm{t}}$ that maximizes their similarity. The objective is

$$\underset{\boldsymbol{v}_i^{\mathrm{t}}}{\operatorname{maximize}} \cos(\boldsymbol{r}_{\boldsymbol{A}^{\mathrm{s}}}(\boldsymbol{v}_i), \boldsymbol{r}_{\boldsymbol{A}^{\mathrm{t}}}(\boldsymbol{v}_i^{\mathrm{t}})). \tag{5}$$

which can be accomplished by gradient descent. This procedure is repeated for $i = 1, \cdots, m$ to obtain all the embeddings of a length-$m$ prompt for the target language model.

It is noted that such searched prompt embeddings may not have the same scale as the target language model's word embeddings, partially because the $\cos$ measure used in (3) and (4) is insensitive to vector magnitude. Therefore, we normalize them as

$$\widetilde{\boldsymbol{v}}_i^{\mathrm{t}} = \frac{\boldsymbol{v}_i^{\mathrm{t}} - \mu_v}{\sigma_v} \cdot \sigma + \mu \tag{6}$$

with $\mu_v, \sigma_v \in \mathbb{R}$ being the mean and standard deviation of all searched prompt embedding values, and $\mu, \sigma \in \mathbb{R}$ being those of pretrained word embeddings of the target model.

The transferred continuous prompt, after the normalization, is directly used to query the target model $P_{\mathrm{t}}(\cdot)$ for inference, given by $P_{\mathrm{t}}(y \mid \widetilde{\boldsymbol{v}}_1^{\mathrm{t}}, \cdots, \widetilde{\boldsymbol{v}}_m^{\mathrm{t}}, x)$.

## 2.4 MULTI-SOURCE TRANSFER

We argue that the induced prompt embeddings from a single model might be model-specific, which is supported by the evidence in Khashabi et al. (2022) that a language model can generate numerous continuous prompts capable of performing the same task. Such flexibility is believed to arise from the high expressiveness of the model's lower layers (Telgarsky, 2016; Raghu et al., 2017). Therefore, the induced continuous prompt from a specific model may be one of the many plausible ones, carrying model-specific information in addition to task semantics and limiting the transferability to other language models.

To enhance the generalization of the induced continuous prompt, we propose a multi-source transfer approach. Specifically, we search for the target embeddings $\boldsymbol{v}^{\text{t}}$ whose encoded embeddings $\boldsymbol{r}_{\boldsymbol{A}^{\text{t}}}(\boldsymbol{v}^{\text{t}})$ align closely with the encoded embeddings $\boldsymbol{r}_{\boldsymbol{A}^{\text{s}_i}}(\boldsymbol{v}^{\text{s}_i})$ from multiple source models $\text{s}_i$ in the relative space. In other words, the goal is to search for $\boldsymbol{v}^{\text{t}}$ such that the sum of similarities between $\boldsymbol{r}_{\boldsymbol{A}^{\text{t}}}(\boldsymbol{v}^{\text{t}})$ and each source prompt $\boldsymbol{r}_{\boldsymbol{A}^{\text{s}_i}}(\boldsymbol{v}^{\text{s}_i})$ is maximized. Given $S$-many source models, the objective of searching a target embedding $\boldsymbol{v}_i^{\text{t}}$ is:

$$\underset{\boldsymbol{v}_i^{\text{t}}}{\text{maximize}} \sum_{j=1}^{S} \cos(\boldsymbol{r}_{\boldsymbol{A}^{\text{s}_j}}(\boldsymbol{v}_i^{\text{s}_j}), \boldsymbol{r}_{\boldsymbol{A}^{\text{t}}}(\boldsymbol{v}_i^{\text{t}})). \tag{7}$$

We follow Eqn. (6) to normalize $\boldsymbol{v}_i^{\text{t}}$ to target model's embedding space, and use the resulting vectors $\widetilde{\boldsymbol{v}}_i^{\text{t}}$ for inference.

# 3 EXPERIMENTS

## 3.1 DATASET

We utilized a widely used factual probing dataset, LAMA (Petroni et al., 2019), to evaluate the effectiveness of our continuous prompt transfer approach. We followed recent factual probing studies (Shin et al., 2020; Zhong et al., 2021) that focus on the TREx split of LAMA. Specifically, LAMA-TREx presents a factual knowledge piece as a triple ⟨subject, relation, object⟩. For example, the fact that "Dante was born in Florence" is represented as ⟨Dante, place of birth, Florence⟩, where "place of birth" is a pre-defined relation in the dataset. In total, there are 41 distinct relations as subtasks, each of which contains up to $1,000$ tuples. We chose factual probing for two key reasons: First, induced prompts represent different task semantics from 41 sub-tasks (distinct pre-defined relations), providing a robust way to evaluate the generalizability of our transfer approach in various scenarios. Second, the factual probing task requires the model to precisely predict the correct entities from its vocabulary. This makes it easier to judge the performance of a prompt.

On the source pretrained language model, we adopted OptiPrompt (Zhong et al., 2021) for inducing a continuous prompt for each of the 41 sub-tasks. Given a sub-task of a certain relation, the source language model is queried using the prompt defined in Eqn. (1), where the prompt embeddings are randomly initialized and then optimized according to Eqn. (2).

## 3.2 CHOICE OF MODELS AND OTHER IMPLEMENTATION DETAILS

We investigated our transferring approach across a range of language models, namely, BERT (Devlin et al., 2019), RoBERTa (Liu et al., 2019), and ALBERT (Lan et al., 2020), including base and large variants. It should be noted that ALBERT utilizes parameter sharing across layers and reduced embedding dimensions, which, to some extent, ties the semantics of embeddings and hidden layers. Therefore, we only used BERT and RoBERTa as the source language models while excluding ALBERTA due to its unique architecture that does not support full compatibility with BERT and RoBERTa. All these models are considered as target models for transfer.

In the main experiments, we set the default number of prompt embeddings $m$ to 5, and the number of anchors $k$ to 8192. We report the standard evaluation metric, micro-average accuracy, which is calculated by averaging the accuracy of 41 sub-tasks of distinct relations in the LAMA dataset (Shin et al., 2020; Zhong et al., 2021). These settings are applied to both our approach and baselines. More implementation details are shown in Appendix A.1.

## 3.3 MAIN RESULTS

**Direct transfer.** We first analyze a naïve method, direct transfer, which directly applies the source-induced continuous prompt to the target model. This provides us with a general understanding of whether continuous prompts are directly transferable. We show the results of direct transfer in Table 1 as a standalone experiment, as it does not fit our main table because direct transfer is only feasible when the source and target embedding dimensions match.

The results reveal that continuous prompts induced from the base models of BERT and RoBERTa (both with 768 dimensions) perform poorly when transferred to each other (around $0.1\%$ accuracy). For their large variants, the transfer performance from RoBERTa to BERT improves marginally, achieving around $0.5\%$ accuracy. Transfer-

Table 1: Accuracy of direct transfer between models with the same embedding dimension. Results are in percentage.

| Source | Target | $d_{\text{embedding}}$ | Transfer acc (%) |
|---|---|---|---|
| $\text{BERT}_{\text{base}}$ | $\text{RoBERTa}_{\text{base}}$ | 768 | 0.11 |
| $\text{RoBERTa}_{\text{base}}$ | $\text{BERT}_{\text{base}}$ | 768 | 0.12 |
| $\text{BERT}_{\text{large}}$ | $\text{RoBERTa}_{\text{large}}$ | 1024 | 6.27 |
| $\text{RoBERTa}_{\text{large}}$ | $\text{BERT}_{\text{large}}$ | 1024 | 0.49 |

ring from BERT to RoBERTa achieves nearly $6.3\%$ accuracy, but is still far from ideal. Overall, this experiment verifies that continuous prompts are not directly transferable, signifying the importance of prompt transfer research.

**Baselines and single-source transfer.** Table 2 represents the results of non-transfer baselines for reference. In the **random** method, prompt embeddings are randomly sampled from a normal distribution fitted to the target models' word embeddings. Its all-zero performance implies that factual probing is a challenging task that requires non-trivial efforts from a machine learning model. In **direct tuning**, the continuous prompt is directly tuned with the target model. As expected, direct tuning achieves high performance, but is undesired as it requires backpropagation through the target model; it serves as an "upper bound" of prompt transfer in the factual probing task. We also present the performance of **manual** prompts, provided by the LAMA dataset (Petroni et al., 2019), serving as another reference score for evaluating prompt transfer methods.

Table 3 shows the main results of our proposed method along with several continuous prompt transfer baselines. We experimented with a straightforward method, called **discretization** (Khashabi et al., 2022), for continuous prompt transfer. Specifically, each prompt embedding is projected to its nearest-neighbor token embedding, and these discrete tokens are transferred to the target model. As analyzed in Khashabi et al. (2022), such a method yields poor transferability, probably due to the

Table 2: Results of the non-transfer baselines, serving as reference scores for transfer.

| Target
Method | $\text{BERT}_{\text{base}}$ | $\text{BERT}_{\text{large}}$ | $\text{RoBERTa}_{\text{base}}$ | $\text{RoBERTa}_{\text{large}}$ | $\text{ALBERT}_{\text{base}}$ | $\text{ALBERT}_{\text{large}}$ |
|---|---|---|---|---|---|---|
| Random | 0.00 | 0.00 | 0.00 | 0.00 | 0.00 | 0.00 |
| Manual | 30.64 | 32.22 | 20.48 | 23.59 | 18.63 | 24.44 |
| Direct tuning | 50.56 | 51.97 | 46.24 | 41.06 | 42.98 | 43.78 |

Table 3: Main results. Best performance is highlighted in **bold**, while second-best performance is underlined. The numbers in gray are the self-transfer performance.

| Target
Source | $\text{BERT}_{\text{base}}$ | $\text{BERT}_{\text{large}}$ | $\text{RoBERTa}_{\text{base}}$ | $\text{RoBERTa}_{\text{large}}$ | $\text{ALBERT}_{\text{base}}$ | $\text{ALBERT}_{\text{large}}$ |
|---|---|---|---|---|---|---|
| **Discretization** | | | | | | |
| $\text{BERT}_{\text{base}}$ | 12.93 | 10.76 | 10.88 | 11.96 | 11.44 | 11.10 |
| $\text{RoBERTa}_{\text{base}}$ | 12.31 | 10.35 | 13.51 | 11.01 | 11.67 | 12.33 |
| $\text{BERT}_{\text{large}}$ | 9.00 | 11.02 | 5.64 | 11.93 | 7.35 | 6.66 |
| $\text{RoBERTa}_{\text{large}}$ | 1.35 | 0.70 | 2.28 | 6.22 | 3.15 | 2.64 |
| **Neural projector** | | | | | | |
| $\text{BERT}_{\text{base}}$ | 26.82 | 12.49 | 14.36 | 9.78 | 10.99 | 18.77 |
| $\text{RoBERTa}_{\text{base}}$ | 23.46 | 17.46 | 35.37 | 20.16 | 11.63 | 14.44 |
| $\text{BERT}_{\text{large}}$ | 3.15 | 4.77 | 5.64 | 4.66 | 8.18 | 14.55 |
| $\text{RoBERTa}_{\text{large}}$ | 2.62 | 3.20 | 5.54 | 12.80 | 7.45 | 8.25 |
| **Single source (ours)** | | | | | | |
| $\text{BERT}_{\text{base}}$ | 49.82 | 31.40 | 17.68 | 21.07 | 20.83 | 16.80 |
| $\text{RoBERTa}_{\text{base}}$ | **31.33** | 27.52 | 45.17 | 25.09 | 26.11 | 24.72 |
| $\text{BERT}_{\text{large}}$ | 26.78 | 50.21 | 7.64 | 16.91 | 15.10 | 13.44 |
| $\text{RoBERTa}_{\text{large}}$ | 3.81 | 12.45 | 3.63 | 40.91 | 4.48 | 2.94 |
| **Dual sources (ours)** | | | | | | |
| $\text{BERT}_{\text{base}} + \text{BERT}_{\text{large}}$ | 49.21 | 47.78 | **27.60** | 23.21 | 23.67 | 22.32 |
| $\text{BERT}_{\text{base}} + \text{RoBERTa}_{\text{base}}$ | 48.79 | **32.83** | 43.83 | **25.26** | **27.13** | **26.54** |

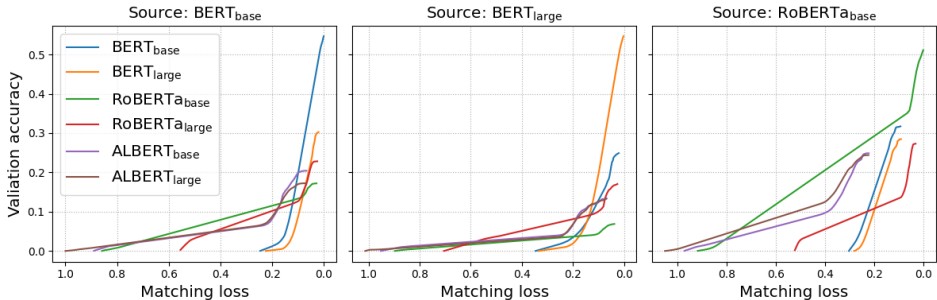

Figure 2: Validation accuracy vs. matching loss, with the curves showing the performance of various target models.

expressive power in the neighbor of discrete token embeddings. Our results also demonstrate the low performance of discretization, which is consistent with previous work and indicates that a more nuanced approach is needed for effective prompt transfer.

In addition, we included an intuitive baseline method, the **neural projector** (Su et al., 2022), for comparison. Specifically, we first trained a two-layer projector to map the source embedding space to the target one based on anchor words. Then, we projected the source-induced prompt embeddings to the target model using the trained projector. Detailed settings and results are provided in Appendix A.2. As seen, transferring the induced prompt embeddings through the projector provides better results but still falls short of manual prompting.

Now, we consider our proposed prompt transfer method with a **single source**. As seen, our method yields consistent improvement compared with the neural projector, which is a compelling result as our method does not require learning a mapping from the source embedding space to the target. This verifies that our proposal of working with a relative space is more effective than the original embedding space for continuous prompt transfer. More profoundly, the prompts transferred from the base models of BERT and RoBERTa surpass the manual prompting baseline, manifesting the practical value of our proposed prompt transfer method.

**Multi-source transfer.** Finally, we evaluate our proposed multi-source prompt transfer method as described in Section 2.4. We consider two **dual-source** settings: BERT$_{base}$+BERT$_{large}$ and BERT$_{base}$+RoBERTa$_{base}$. The results are also shown in Table 3. As seen, using multiple sources generally improves transferability. For example, the BERT$_{base}$+BERT$_{large}$ dual-source setting outperforms BERT$_{base}$ by 2–10 percentage points, although BERT$_{large}$ alone is not a strong source model. Compared with the best single source, the BERT$_{base}$+RoBERTa$_{base}$ dual-source setting yields an improvement of 1–2 points on the target models of ALBERT$_{base}$ and ALBERT$_{large}$, which are not involved in the prompt tuning process. Overall, this experiment verifies that multiple sources improve the transferability of continuous prompts.

**Transferability and expressive power.** We observed in Table 3 that a larger source model (either BERT$_{large}$ or RoBERTa$_{large}$) has lower transfer performance. This aligns with the intuition in Khashabi et al. (2022) that there could be a large number of similarly performing continuous prompts, residing in a large (and also deep) model's expressive embedding space (Telgarsky, 2016; Raghu et al., 2017). Therefore, the induced prompt may carry model specificity in addition to task semantics, limiting its transferability to other models.

Fortunately, the low transferability of large source models does not affect the value of our work, because our typical application scenario is to tune a continuous prompt on a small source model and use it for a large model. This is precisely the setting where our approach is especially effective.

### 3.4 ANALYSIS

**Matching in relative space vs. transferability.** Our approach is based on the intuition that the task semantic is carried in a relative space, which can be aligned for different language models. We analyze whether a closer matching of the relative space yields a higher transferability of the continuous prompts. In Figure 2, we show the trend of validation accuracy versus matching loss along the search process in Eqn. (5), where for each source–target combination, we averaged the performance of all sub-tasks.

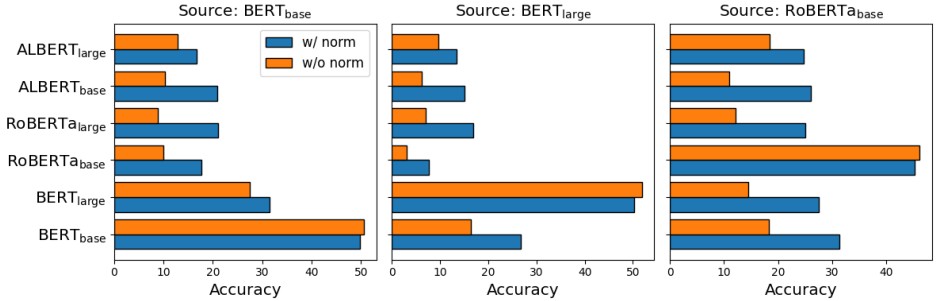

Figure 3: The effect of normalization.

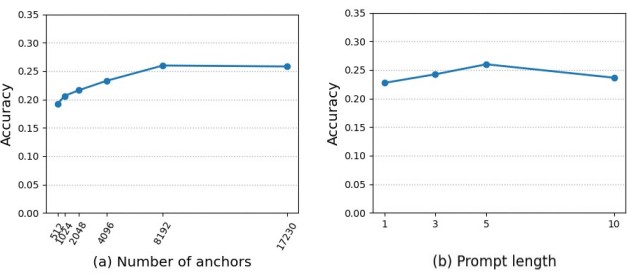

Figure 4: The effect of the anchor number and prompt length. Each value (dot) was computed by averaging the accuracy from all source–target combinations.

In Figure 2, we observe that, as the matching loss decreases (towards right in the plots), the validation accuracy of target models increases. In the special case where source and target models are identical, the matching loss of the relative space is able to approach zero, and the accuracy of the transferred prompt is close to that of the source prompt. This demonstrates that our approach is able to recover the original embeddings from the relative space, even if a normalization is performed to the target embeddings in Eqn. (6).

When source and target models are different, the matching loss does not approach zero, which is reasonable because the different language models' embedding spaces may not be perfectly aligned. Nevertheless, the correlation between matching loss and validation accuracy is highly consistent across all source–target combinations, convincingly showing that a better matching in the relative space leads to a more transferable continuous prompt.

**Effect of normalization to target embedding space.** We further provide an ablation study on the normalization of target embeddings introduced in Eqn. (6). Specifically, we compare the performance of the target prompts with or without the normalization treatment.

From Figure 3, we observe that, when the source and target models are identical (non-transfer settings), the normalization hurts the performance, as it distorts the original word embedding space. However, the gap is minimal, which provides additional evidence that we are able to recover the original embeddings through our relative space even with the normalization.

When the source and target models are different, however, the normalization significantly improves the performance. The results confirm that the normalization treatment can better cast the relative embedding of a source-induced continuous prompt into the target embedding space, directly understood by the target language model.

**Effect of the anchor numbers and prompt length.** We analyze the number of anchors and the prompt length. We first varied the number of anchors from the set $\{512, 1024, 2048, 4096, 17230\}$, where $17,230$ is the number of the shared tokens in different language models considered in this study. The anchor number decides the feature dimensionality of the relative representations, shown in Eqns. (3) and (4). Figure 4a reveals a general trend of improved transfer performance with an increasing number of anchors. When we have $512$ anchors, the transfer performance is the lowest, which is due to the inadequate capacity of the low-dimensional relative space. On the other hand, using the entire shared vocabulary results in a marginal decrease in performance. This is reasonable because the embeddings of rare words may not be well trained, consequently introducing noise to the high-dimensional feature space.

We then investigate the effect of prompt length on transfer performance, where we chose the length from the set $\{1, 3, 5, 10\}$. We observe in Figure 4b that the performance of transfer improves until a certain point, namely, five virtual tokens in our case. With a prompt length of 10, the transfer performance decreases slightly.

Overall, our approach is robust to these hyperparameters. Based on this analysis, we set the number of anchors to 8192 and the prompt length to 5 in our main experiments (see § 3.2).

**Additional results.** We provide additional results in the appendices. These include a study of transferring induced prompts across different model architectures, such as from BERT to GPT-2, detailed in §B.1. Furthermore, we demonstrate the applicability of our method to classification tasks, discussed in §B.2.

## 4 RELATED WORK

In recent years, language models (LMs) have shown impressive few-shot learning capabilities that allow them to be adapted to a variety of downstream tasks through the design of textual prompts (Brown et al., 2020). Subsequent research improves the performance of NLP tasks by creating discrete prompts that are manually crafted, searched through gradient descent (Shin et al., 2020), or using reinforcement learning (Deng et al., 2022). Meanwhile, there has been a growing interest in continuous prompts tuning (Li & Liang, 2021; Lester et al., 2021; Zhong et al., 2021). These studies suggest that tuning a small number of parameters in prompt embeddings can match the performance of full-model finetuning (Houlsby et al., 2019; Lester et al., 2021), which shows the potential of continuous prompt tuning.

Several previous studies have tried to utilize the induced continuous prompt to other tasks or other LMs. For example, Vu et al. (2022) show that prompt embeddings induced on a certain task can be used to initialize the prompts for similar tasks. This led to further research on retrieving and mixing continuous prompts for new tasks (Asai et al., 2022; Su et al., 2022; Wang et al., 2023). Su et al. (2022) further study the transferability of continuous prompts in cross-LM scenarios. They propose to train a projector between the embedding space of two LMs with parallel induced prompt embeddings or task signals, which contrasts with the zero-shot transfer approach in this work.

Rakotonirina et al. (2023) and Wen et al. (2023) investigate zero-shot transferability between different LMs using the induced discrete prompts. Their work is orthogonal to ours as we focus on the cross-model transfer of continuous prompts. Although transferring discrete prompts offers greater simplicity compared with our proposed continuous prompt transfer, continuous prompts are more versatile and can be adapted to a broader range of applications.

The concept of mapping different embeddings into a shared latent space has been well explored in the cross-lingual scenario (Faruqui & Dyer, 2014; Lazaridou et al., 2015; Artetxe et al., 2018), which further paves the way for unsupervised neural machine translation (Lample et al., 2018). We follow the assumption from these studies and assume the induced task embeddings (Vu et al., 2022) from different language models share similar latent structures. We employ relative representation (Moschella et al., 2023) to encode a source-induced prompt, and decode it to the embedding space of the target model.

## 5 CONCLUSION

We introduced a zero-shot method for transferring continuous prompts between different language models through a relative space. Experiments confirm the effectiveness of our approach, and we further provide insights into the correlation between a model's transferability and its expressive power. Moreover, we propose a simple method to improve the generalizability of prompt transfer by using multiple source models.

Given the shift towards unified large language models (Brown et al., 2020), our method could enable smaller source models to act as effective "soft prompt engineers" that perform better than manual prompting. Additionally, it is a promising direction to explore direct human–model interactions that bypass the need for discrete language. This will involve prompting pretrained language models using continuous prompts transferred from sources like encoded brain signals (Zou et al., 2021).

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

Table 4: Details of the pretrained language models considered in this study. MLM, NSP, SOP, and NTP stand for masked language modeling, next sentence prediction, sentence order prediction, and next token prediction, respectively. It should be noted that ALBERT employs weight sharing, and its memory consumption is similar to BERT and RoBERTa.

| Model | | #Parameters | $d_{\text{hidden}}$ | $d_{\text{embedding}}$ | Pretraining task | Pretraining data |
|---|---|---|---|---|---|---|
| BERT | base | 110M | 768 | 768 | MLM & NSP | BookCorpus, English Wikipedia |
| | large | 340M | 1024 | 1024 | | |
| RoBERTa | base | 125M | 768 | 768 | MLM | BookCorpus, English Wikipedia, CC-News, OpenWebText, Stories |
| | large | 355M | 1024 | 1024 | | |
| ALBERT | base | 12M | 768 | 128 | MLM & SOP | BookCorpus, English Wikipedia |
| | large | 18M | 1024 | 128 | | |
| GPT-2 | small | 117M | 768 | 768 | NTP | WebText |
| | base | 345M | 1024 | 1024 | | |
| | large | 774M | 1280 | 1280 | | |

## A  IMPLEMENTATION DETAILS

### A.1  DETAILS OF THE MODELS

Table 4 provides an overview of the language models used in this study, including base and large variants of BERT, RoBERTa, and ALBERT. Each model is trained with distinct pretraining tasks and datasets. In this study, we focus on transferring continuous prompts between masked language models, as this fill-in-the-blank mechanism is a natural way to probe knowledge (Shin et al., 2020). We also provide a preliminary empirical investigation of transferring continuous prompts between different model structures, e.g., from the encoder-only BERT model to the decoder-only GPT-2 model, which is discussed in §B.1.

Due to the variations in pretraining datasets and tokenizing methods, the language models in different families (e.g., BERT vs. RoBERTa) have different vocabularies. We obtained a shared vocabulary of tokens by taking the intersection of these individual vocabularies. During the transfer, we first encode the source prompt embeddings to the entire relative space. Then, we pick top-$k$ dimensions of highest values ($k = 8192$) and set the rest of zero, which follows Norelli et al. (2022).

### A.2  DETAILS OF THE PROJECTOR BASELINE

One of our baselines is a projector that maps the source embedding space to the target one. We trained a two-layer neural network as the projector based on the shared vocabulary. Specifically, we have

$$\text{Proj}(\boldsymbol{e}_i^{\text{s}}) = W_2(f(W_1\boldsymbol{e}_i^{\text{s}} + \boldsymbol{b}_1)) + \boldsymbol{b}_2, \tag{8}$$

where $f$ is the Leaky ReLU activation function (Xu et al., 2015). For some anchor word $i$, we denote by $\boldsymbol{e}_i^{\text{s}}$ and $\boldsymbol{e}_i^{\text{t}}$ the word embeddings of the source model and target model, respectively. We train the projector by minimizing the mean squared error loss:

$$\mathcal{L}_{\text{MSE}} = \frac{1}{k} \sum_{i=1}^{k} (\text{Proj}(\boldsymbol{e}_i^{\text{s}}) - \boldsymbol{e}_i^{\text{t}}), \tag{9}$$

where $k$ is the size of shared vocabulary between two language models. We trained the neural network with 10 epochs using the Adam optimizer (Kingma & Ba, 2014). The learning rate was 5e-3 and the batch size was 16. The hidden dimension of this two-layer neural network was 768. We ran the validation on target models after each training epoch with the projected target prompt embeddings. We chose the projector with the highest validation performance and used it for test.

## B  ADDITIONAL RESULTS

In this appendix, we report preliminary results of the additional experiments conducted during the author response phase based on the reviewers' suggestions. In particular, we show the adaptability

Table 5: Results on transferring prompts between encoder and decoder models.

| Method \ Target | | BERT$_{base}$ | RoBERTa$_{base}$ | GPT2$_{small}$ | GPT2$_{medium}$ | GPT2$_{large}$ |
|---|---|---|---|---|---|---|
| Direct tuning | | 50.56 | 46.24 | 31.62 | 32.23 | 34.44 |
| Manual | | 30.64 | 20.48 | 4.73 | 8.01 | 10.23 |
| Source | BERT$_{base}$ | - | 17.68 | 10.46 | 11.52 | 5.50 |
| | RoBERTa$_{base}$ | 31.33 | - | 14.06 | 13.70 | 14.33 |
| | GPT2$_{small}$ | 6.58 | 0.39 | - | 13.72 | 2.34 |
| | GPT2$_{medium}$ | 4.06 | 0.50 | 5.02 | - | 1.79 |

Table 6: Results of transferring prompts from source models to RoBERTa$_{large}$ on the SST-2 and DPpedia classification tasks.

| Method | SST-2 (accuracy) | DBpedia (accuracy) |
|---|---|---|
| Direct tuning | 90.94 | 84.92 |
| Manual | 69.95 | 72.28 |
| Source: BERT$_{base}$ | 82.45 | 77.05 |
| Source: RoBERTa$_{base}$ | 84.63 | 80.81 |

of our method to different model architectures in §B.1, and experiment with classification tasks in §B.2.

## B.1 TRANSFER BETWEEN DIFFERENT MODEL ARCHITECTURES

We first demonstrate the feasibility of transferring continuous prompts across different model architectures. This experiment explores the transferability between encoder and decoder models, focusing on generative GPT-2 models of varying sizes: small, medium, and large, as detailed in Table 4. We selected BERT$_{base}$ and RoBERTa$_{base}$, two encoder models, for our primary experiment to examine the transferability of prompts to or from GPT-2 models.

Table 5 shows the results of transferring continuous prompts across architectures on the LAMA dataset, including comparisons with the performance of directly tuned and manually prompted target models for reference. We see that the prompts induced on the encoder models, BERT$_{base}$ and RoBERTa$_{base}$, are transferable to the GPT-2 models with different sizes. Notably, RoBERTa$_{base}$ shows its best transferability, outperforming the manual prompting baseline across all target models. However, we found that the GPT-2 models as the source cannot induce as meaningful prompts as the encoder models, often underperforming manual prompting. The underlying reason contributing to the poor transferability of the continuous prompts induced on GPT-2 models remains unexplored and merits further study.

## B.2 RESULTS ON CLASSIFICATION TASKS

Now we show our proposed transfer method is effective on other NLP tasks. Specifically, we include SST-2, a binary sentiment classification task, and DBpedia, a 14-category topic classification task. Unlike LAMA's entity prediction which requires the model to consider the whole vocabulary, the classification task only requires prediction within the label words based on the prompt, for example, "great" or "bad" for the SST-2 dataset (Sun et al., 2022).

As shown in Table 6, compared to using manual prompts on the target model directly, transferring prompts from both BERT$_{base}$ and RoBERTa$_{base}$ to the RoBERTa$_{large}$ target model yields better results. In line with our previous findings, RoBERTa$_{base}$ shows its superior transferability. Overall, our additional results present the potential of applying our approach to various tasks and model architectures.

ACKNOWLEDGMENTS

We would like to thank Ning Shi for his insightful suggestion of the multi-source transfer approach. We also thank all reviewers and chairs for their valuable and constructive comments. The research is supported in part by the Natural Sciences and Engineering Research Council of Canada (NSERC) under Grant No. RGPIN2020-04465, the Amii Fellow Program, the Canada CIFAR AI Chair Program, a UAHJIC project, an Alberta Innovates Program, and the Digital Research Alliance of Canada (alliancecan.ca).

