# OpenReview forum: "Zero-Shot Continuous Prompt Transfer: Generalizing Task Semantics Across Language Models"
_ICLR.cc/2024/Conference — ICLR 2024 poster_

### Official Review · Reviewer_AxcL · 2023-10-28

**Soundness:** 3 good
**Presentation:** 3 good
**Contribution:** 3 good
**Rating:** 6
**Confidence:** 4

**Summary:**

The authors propose a new method for zero-shot continuous prompt transfer from a pretrained language models to another. This method firstly finds the anchors between two language models (subsets of shared vocabulary) to construct a relative space. Then source prompts are encoded in the relative space and corresponding prompts in the target space are searched based on the source embeddings. Extensive experiments and analysis are carried out to support the effectiveness of this method.

**Strengths:**

1. The motivation is strong and clear. Continuous prompt tuning is an important issue, and this method allows easier transfer between model-specific continuous prompts. This may contribute to further research of parameter efficient learning and continual learning, etc.

2. The idea of using vocabulary as anchors and connectors between models is novel and interesting.

**Weaknesses:**

1. Method limitation: It seems that this method highly relied on shared vocabularies. Quality of vocabulary and selection of shared tokens may be important but no analysis is given.

2. Clarification: Some key details are not clarified (See questions)

3. Minor: Inconsistent use of search loss and matching loss in Section 3.4.

**Questions:**

1. When “shared vocabulary” is mentioned in the fourth paragraph of the Introduction Section, does it mean the vocabularies should be the same or just have some common parts?
2. In Equations 5 and 7, what is the optimization method and how do you implement this?
3. In experiments, BERT based models are deliberately chosen. Will this method be applicable to generative models such as GPT? This question is especially important when LLMs have recently been popular because one can use this method to find a set of optimal prompts on a small model and transfer them to a large one.
4. How are the anchors chosen? Are they randomly sampled? Will the selection process affect model performance?

---

> ### Author Response · Authors · 2023-11-16
> **Response to Reviewer AxcL**
>
> We thank the reviewer for recognizing the contribution and novelty of our work.
>
> >Weaknesses 1 & Question 4: “Quality of vocabulary and selection of shared tokens may be important but no analysis is given”. “ How are the anchors chosen? Are they randomly sampled? Will the selection process affect model performance?”
>
> Thanks for pointing out the importance of the shared vocabulary because it provides the ground for prompt transfer. However, we would like to mention that the performance of the transfer is not sensitive to the number of anchors. In Figure 4, we show that even if we use only 512 anchors, there is not much decrease in transfer performance. For the selection of anchors, Appendix A.1 shows that we may simply pick the top-k nearest neighbors based on the prompt embeddings, and we applied this to all the experiments for our method.
>
> >Weakness 2: “Inconsistent use of search loss and matching loss in Section 3.4.”
>
> Thanks for pointing out the inconsistency of terminologies. We fixed them in the paper.
>
> >Question 1: “Does it mean the vocabularies should be the same or just have some common parts?”
>
> We only require the vocabularies to share a common subset, which is simply the intersection of the BERT, RoBERTa, and ALBERT’s vocabularies in our experiments.
>
> >Question 2: “In Equations 5 and 7, what is the optimization method and how do you implement this?”
>
> We used gradient descent. Clarified after Eqn 5. Thanks!
>
> >Question 3: “Will this method be applicable to generative models such as GPT?”
>
> Thanks for pointing out this interesting further direction. We didn’t include generative models because we adhered to the framework established in the OptiPrompt paper, utilizing identical models and the factual probing dataset. We’ve now extended OptiPrompt and tried generative models, namely, gpt2. The results are as follows.
>
> | source \ target | gpt2-small | gpt2-medium |
> |---|---|---|
> | direct tune   | 31.62      | 32.23      |
> | manual        | 4.62       | 8.98        |
> | gpt2-small    | -          | 13.72       |
> | gpt2-medium   | 5.02       | -           |
> | bert-base     | 10.46      | 11.52       |
> | roberta-base  | 14.06      | 13.70       |
>
> We found that our method is indeed applicable to generative models, and transferring the induced prompts from masked language models to generative models is also possible.

---

> > ### Comment · Reviewer_AxcL · 2023-11-21
> >
> > Thanks for your response. It seems that the performance of "manual" is extremely low. Do you have any comments? Thanks!

---

> > > ### Author Response · Authors · 2023-11-21
> > > **Response to Reviewer AxcL**
> > >
> > > We appreciate your observation regarding the low performance of GPT2 models with manual prompts. We hypothesize that there are a couple of factors potentially contributing to this observation:
> > >
> > > **Generative model's training:**
> > >  Both small and medium GPT2 models are trained to predict the next word in a sequence and are optimized for language coherence. This training approach often leads to a phenomenon known as "hallucination", where the model prioritizes fluent next word over factual accuracy.
> > >
> > > **Nature of Manual Prompts:**
> > > The manual prompts provided by the dataset are not always conducive to the GPT2's prediction style. Of the 41 prompts for different relations, 3 particularly require contextual understanding for entity prediction. For example, the effectiveness of the manual prompt for music genre ("[X] plays [mask] music") may rely on the model recognizing "music" as the context. In GPT2's case, when processing the input prefix "[X] plays", it lacks awareness of the subsequent "music" context, leading to less accurate predictions. This contrasts with masked language models like BERT or RoBERTa, which process the entire prompt more holistically.

---

> > > > ### Comment · Reviewer_AxcL · 2023-11-22
> > > >
> > > > I believe using prompts like "[X] plays [mask] music" directly is not suitable. I encourage you to rewrite them to fit generative models' character and rerun the "manual" experiments.

---

> > > > > ### Author Response · Authors · 2023-11-22
> > > > > **Response to Reviewer AxcL**
> > > > >
> > > > > Thanks for the suggestion! After taking a closer look at the LAMA-provided manual prompts, we found that 6 out of 41 prompts do not have the [mask] token at the end. And we rewrote those prompts as follows.
> > > > > 1. [X] plays in [mask] position -> [X] plays in the position of
> > > > > 2. [X] is a [mask] by profession  -> [X] is a professional
> > > > > 3. [X] is [mask] citizen  -> [X] is a citizen of
> > > > > 4. [X] plays [mask] music -> [X] plays the music genre of
> > > > > 5. [X] is affiliated with the [mask] religion -> [X] is affiliated with the religion of
> > > > > 6. [X] and [mask] are twin cities -> The twin city of [X] is
> > > > >
> > > > > We then applied the revised set of manual prompts for model inference.
> > > > > | source \ target | gpt2-small | gpt2-medium |
> > > > > |---|---|---|
> > > > > | original manual | 4.62  | 8.98   |
> > > > > | revised manual  | 4.73    |  8.01  |
> > > > >
> > > > > We found that there is a slight increase in performance for the gpt2 small model, but the performance of the gpt2 medium model has a slight decrease. This shows that using manual prompts may not be as effective for factual probing the gpt2 models. Nevertheless, the design of manual prompts does not hurt the contribution of our method, as we show the transferability of the induced soft prompt from one model to the other.

---

> > > > > > ### Comment · Reviewer_AxcL · 2023-11-22
> > > > > >
> > > > > > Thank you for your response. The idea is interesting, but given the overall performance, I have decided to maintain my current overall rating.

---

> > > > > > > ### Author Response · Authors · 2023-11-22
> > > > > > > **Response to Reviewer AxcL**
> > > > > > >
> > > > > > > Thank you for your time and your recognition of the idea as interesting. We are motivated to further pursue this direction in our future work.

---

### Official Review · Reviewer_NAhq · 2023-10-30

**Soundness:** 3 good
**Presentation:** 3 good
**Contribution:** 2 fair
**Rating:** 6
**Confidence:** 4

**Summary:**

The paper presents a zero-shot continuous prompt transfer approach that learns prompt representations for target models (large models) from the representations of source models (small models). The assumption is that target models and source models share some common words as anchors. The learning approach is then forcing the projection of the soft prompts on the source anchors and the projection of those on the target anchors to be similar enough. Evaluation results on factual probing verifies the effectiveness of the approach.

The paper can be improved if the authors could

(1) Compare the proposed method with a straightforward baseline:  v^t_i=\sum_{l=1}^k cos(v^s_i, a^s_l) a^s_l. This is a simplification of the proposed method with no need of learning.

(2) Evaluate the proposed method on tasks other than factual probing. Though there are 14 types of relations in the task, the task itself lacks diversity. The results can be more convincing if the authors could report comparison results on tasks such as text classification, NLI, semantic matching, and QA.

**Strengths:**

a new zero-shot prompt transfer approach
empirical studies on the benchmark of factual probing

**Weaknesses:**

The empirical results are not convincing enough due to the selection of the evaluation task

**Questions:**

N/A

---

> ### Author Response · Authors · 2023-11-16
> **Response to Reviewer NAhq**
>
> We thank the reviewer for the comments.
>
> >Weakness 1 “Compare the proposed method with a straightforward baseline: v^t_i=\sum_{l=1}^k cos(v^s_i, a^s_l) a^s_l”:
>
> We believe there is a minor typo in the formula. If we understand correctly, the suggested method is to set “v^t_i=\sum_{l=1}^k cos(v^s_i, a^s_l) a^t_l”. Thanks for the suggested baseline (which we call weighted target anchors below). We conducted a preliminary study by transferring the induced 41 prompts from BERT-base to BERT-large. The result compared with other methods is as follows.
>
> | Method | Accuracy |
> |---|---|
> |Weighted target anchors |1.95 |
> |Discretization | 10.76 |
> |Neural projector | 12.49 |
> |Single source | 31.40 |
>
> We see that the proposed baseline does not perform very well in our scenario, probably because it weighs target anchors based on the source cosine similarity in a heuristic manner.
>
> >Weakness 2: “Evaluate the proposed method on tasks other than factual probing”
>
> Our choice of factual probing was driven by its suitability for effectively testing prompt transfer across various models, considering the range of subtasks and the inherent difficulty of the task.
>
> We did preliminary experiments on text classification using the SST-2 (2-way sentiment classification) and DBpedia (14-way topic classification) datasets. We keep our default settings and tune five prompt embeddings on two smaller source models (bert and roberta base models) and transfer them to the roberta-large model. The results are as follows.
>
> Performance when transferring to roberta-large
> | source | SST- 2 (accuracy) | DBpedia (accuracy) |
> | --- | --- | --- |
> | manual | 69.95 | 72.28 |
> | bert-base | 82.45 | 77.05 |
> | roberta-base | 84.63 | 80.81 |
>
> We found the transferred prompts' performance is better than manual prompting, which is consistent with the factual probing task.
>
> We hope our response has addressed your concerns, and we believe that our work makes a valuable contribution showing that induced prompt embedding is transferable across different models. Thanks!

---

> > ### Comment · Reviewer_NAhq · 2023-11-20
> > **Acknowledgement**
> >
> > I read the response and find my concerns have been addressed. Therefore, I adjusted my ratings.
> >
> > Thanks for the effort and pointing out my typos.

---

> > > ### Author Response · Authors · 2023-11-21
> > > **Follow-up message to Reviewer NAhq**
> > >
> > > We are glad our response addressed your concerns. Thank you for your reevaluation and raising the score!

---

### Official Review · Reviewer_2xK8 · 2023-10-31

**Soundness:** 3 good
**Presentation:** 3 good
**Contribution:** 3 good
**Rating:** 8
**Confidence:** 3

**Summary:**

The paper explores methods to train continuous/soft prompts in a source model and then transfer that prompt (and use it directly) for a target model. This can be useful to do relatively expensive soft prompt tuning on small source models and transfer the trained prompt to a bigger target model (compared to training the big model itself).

The transfer idea involves encoding soft prompts from the source in a "relative space" (encode cosine similarity scores between soft prompt tokens and selected source token embeddings - "anchors") and then trying to search for target prompts that achieve similar similarity scores against corresponding anchor embeddings of the tokens in the target model. This can be done without backpropagating through the target model. This method can be also extended for multi-source transfer setup.

**Strengths:**

1. The target of transferring soft-prompt task semantics is an underexplored area of study. The target is well motivated in the paper.

2. The method is novel and elegant.

3. The idea works better against some relevant baselines for transferring prompt task semantics from a source model to a target model.

**Weaknesses:**

1. If I understand correctly, at this point the method does not seem practical. We seem to get better accuracy just by directly using the source model. While the current approach transfers better than naive baselines or baselines based on earlier ideas (discretization, projectors), the transfer itself appears like a lose-lose scenario -- because you have to do extra work for transfer, and then you are (generally) trying to run the target soft prompt in a bigger (or same size) model. This seems pointless if just running the base source model gives us better overall performance.

I am willing to accept this despite this because this paper seems like an early foray into the transfer of soft prompts and can inspire future research while serving as a baseline. Please correct me, however, if I am mistaking something about the immediate practical value of transfer given the current method.

**Questions:**

1. How is the optimum of eqn 7 or 5 searched for exactly?

---

> ### Author Response · Authors · 2023-11-16
> **Response to Reviewer 2xK8**
>
> We thank the reviewer for recognizing the novelty of our method by saying “The method is novel and elegant”.
>
> >Weakness (performance lower than the source model):
>
> Thanks for this insight! We acknowledge that currently our transfer performance is lower than the source model, which may hinder its immediate practice. We added a limitation paragraph in the conclusion section discussing this. Nevertheless, our work has certain practical values, as LLMs tend to unify and it’ll be handy if we may just use one LLM to solve various tasks.
>
> As also recognized by the reviewer, our paper addresses an important research direction, and with increasingly powerful target LLMs and improved transfer techniques, we can expect that our research serves as a stepping stone towards more effective transfer methods.
>
> >Question: “How is the optimum of eqn 7 or 5 searched for exactly?”
>
> The search was conducted by gradient descent. In particular, we compute the gradient of the loss (either eqn 5 or eqn 7) wrt to the embeddings of the prompt tokens and update them accordingly. We’ve clarified it in the revision.
>
> It’s also worth mentioning that our code is publicly available (Footnote 1) to support replication.
>
> We thank the reviewer again for the insightful comments and we’ve revised our paper accordingly.

---

> > ### Comment · Reviewer_2xK8 · 2023-11-17
> >
> > Thank you for the feedback. Overall, upon further reflection, revisions, and the additional results I have decided to increase my score to 8 from 6. The idea of unification is interesting and shows the potential of the overall idea of transfer although for immediate practical application, it's still not as clear why one would not just use a good source model for everything given the loss from the transfer. Nevertheless, I am willing to mostly overlook the weakness given many research directions (like neural machine translation) started out without being as immediately practical in the beginning compared to prior methods.

---

> > > ### Author Response · Authors · 2023-11-17
> > > **Follow-up message to Reviewer 2xK8**
> > >
> > > Thank you for your helpful thoughts and for raising the score! We really appreciate it.

---

### Official Review · Reviewer_6csB · 2023-10-31

**Soundness:** 3 good
**Presentation:** 3 good
**Contribution:** 4 excellent
**Rating:** 8
**Confidence:** 2

**Summary:**

This paper proposes a new route to Prompt-Tuning problems. Besides human-writing prompts and doing gradient back-propagation to learn continuous prompt, it brings the possibility of transferring an existing corpus of prompts for different models, regardless of the size of their embedding space.

The idea of the paper involves first translate one model's continuous prompt into a shared high-dimensional space, and search in that space for the target continuous prompt. The experiment shows that prompts are more or less transferable between BERT, RoBERTA, ALBERT, especially if you use dual source prompt transfer.

**Strengths:**

As said by the paper, novelty is a big strength. It's quite unthinkable to transfer learned continuous prompt of one model to another model with different embedding size. And this paper shows that it's possible.

The writing of the paper is clear.

The introduction of the method including translation to shared embedding space, and then search for target continuous prompt makes sense.

I appreciate the experiment design in Table-2, which includes random and manual, as two baselines, along with the learned prompt baseline. The manual baseline(human) is important because it tells me that even though transferred prompt isn't as good as the learned prompt, but it is still competitive with manual prompts.

Overall, I find the idea to be novel, results to be solid (did not beat the learned prompt baseline), but gives overall good performance, compared with human baseline.

**Weaknesses:**

Why is generative models not included for experiments?

I understand the explanation for choosing a factual dataset for evaluation. But the result and the claim by the paper will be much strong if there are more than 1 dataset to support its claim.

**Questions:**

Is BERT embedding transferrable to a GPT2 model?
Or is a GPT2 model embedding transferable to a pythia model?

---

> ### Author Response · Authors · 2023-11-16
> **Response to Reviewer 6csB**
>
> Thank you for your strong support, especially for recognizing the novelty of our work.
>
> >Weakness 1: “Why is generative models not included for experiments?” & Question: “Is BERT embedding transferable to a GPT2 model? Or is a GPT2 model embedding transferable to a pythia model?”
>
> Thank you for the suggestion. We are also interested in the prompt transferability across different models other than the masked language models. We didn’t include generative models because we followed the OptiPrompt paper by using the same settings as they did (same models and the same LAMA dataset).
>
> We’ve now experimented with generative models, and obtained some preliminary results:.
> | source \ target | gpt2-small | gpt2-medium |
> |---|---|---|
> | direct tune   | 31.62      | 32.23       |
> | manual        | 4.62       | 8.98        |
> | gpt2-small    | -          | 13.72       |
> | gpt2-medium   | 5.02       | -           |
> | bert-base     | 10.46      | 11.52       |
> | roberta-base  | 14.06      | 13.70       |
>
> In the experiment, we considered the small and medium versions of the gpt2 model. We found that our method is applicable to generative models as well. Moreover, the transfer from masked language models to generative models is also possible.
>
> >Weakness2: “the result and the claim by the paper will be much strong if there are more than 1 dataset”
>
> Thanks for the suggestion. Again, we followed previous work and used LAMA as the dataset, which contains 41 sub-tasks and provides a comprehensive evaluation for prompting transferability.
>
> We did a preliminary experiment on two classification datasets: SST-2, a 2-way sentiment classification, and DBpedia, a 14-way topic classification. We tuned five soft prompt tokens on BERT and RoBERTa base models, and transferred them to the RoBERTa large model. The results are as follows.
>
> Performance when transferring to RoBERTa-Large
> | source | SST- 2 (accuracy) | DBpedia (accuracy) |
> | --- | --- | --- |
> | manual | 69.95 | 72.28 |
> | bert-base | 82.45 | 77.05 |
> | roberta-base | 84.63 | 80.81 |
>
> We see that the transferred performance is much higher than manual prompting, which shows the clue that our method can be adapted to other datasets and tasks.
>
> We thank the reviewer again for the strong support and valuable suggestions. As our paper on soft prompt transfer is the first of its kind, we’re willing to further pursue this direction in our future work.

---

> > ### Comment · Reviewer_6csB · 2023-11-21
> > **Good experiments**
> >
> > Thank you for uploading new results. They addresses my concerns for the applicability of the method across different architecture.

---

> > > ### Author Response · Authors · 2023-11-22
> > > **Response to Reviewer 6csB**
> > >
> > > We are pleased that our additional experiments have addressed your concern. Thank you for your supportive and insightful feedback!

---

### Meta-Review · Area_Chair_8jLX · 2023-11-30

**Metareview:**

The paper proposes a method to transfer soft prompts across language models by using a second-order "relative representation" space in which prompts are represented in terms of their similarities to pivot words. This is a clever application of the relative-representations methods proposed by Moschella et al at ICLR 2023.

The paper is clearly written and presents an interesting result, obtained with small, somewhat old masked language models (extended, in the rebuttal, to generative models). It's not clear that the result would still be relevant when using very large language models, for which the whole issue of finding effective machine-generated prompts might no longer be relevant. In this respect, the paper might rather be seen as a theoretical contribution to the investigation of generality of language-model representations. However, the paper is somewhat underwhelming in this latter respect, as it doesn't present an analysis of the latent semantics of the discovered prompts.

**Justification For Why Not Higher Score:**

The paper is presenting a clever experiment, but it is clearly rather incremental with respect to the work of Moschella et al, and it is dealing with a problem (optimal soft prompting tuning) that might be somewhat obsolete.

**Justification For Why Not Lower Score:**

The paper IS clever, and it points to a promising way to analyze soft prompt semantics, and thus gain insights on how/why LMs work.

---

### Decision · Program_Chairs · 2024-01-16

Accept (poster)